# Cost-effectiveness evaluation of routine histoplasmosis screening among people living with advanced HIV disease in Latin America and the Caribbean

Radha Rajasingham[1], Narda Medina[2], Gabriel T. Mousquer[3], Diego H. Caceres[4,5], Alexander Jordan[6], Mathieu Nacher[7], Diego R. Falci[8,9], Ayanna Sebro[10], Alessandro C. Pasqualotto[11,12], Omar Sued[13], Tom Chiller[6], Freddy Perez[11,13]*

1 Division of Infectious Diseases & International Medicine, Department of Medicine, University of Minnesota, Minneapolis, Minnesota, United States of America, 2 ASRT, Inc., Mycotic Diseases Branch, Division of Foodborne, Waterborne, and Environmental Diseases (DFWED), Centers for Disease Control and Prevention, Atlanta, Georgia, United States of America, 3 Graduate Program in Biosciences, Federal University of Health Sciences of Porto Alegre (UFCSPA), Porto Alegre, Brazil, 4 Center of Expertise in Mycology Radboudumc/CWZ, Nijmegen, The Netherlands, 5 Studies in Translational Microbiology and Emerging Diseases (MICROS) Research Group, School of Medicine and Health Sciences, Universidad del Rosario, Bogota, Colombia, 6 Mycotic Diseases Branch -Centers for Disease Control and Prevention, Atlanta, Georgia, United States of America, 7 Center for Clinical Investigation Antilles-Guyane, Inserm 1424, Cayenne Hospital, Cayenne, French Guiana, France, 8 Infectious Diseases Unit, Hospital de Clínicas de Porto Alegre, Porto Alegre, Brazil, 9 School of Medicine, Pontifical Catholic University of Rio Grande do Sul, Porto Alegre, Brazil, 10 National AIDS Coordinating Committee Secretariat, Port of Spain, Trinidad & Tobago, 11 Federal University of Health Sciences of Porto Alegre, Porto Alegre, Brazil, 12 Molecular Biology Laboratory, Santa Casa de Misericordia de Porto Alegre, Porto Alegre, Brazil, 13 Communicable Diseases Prevention, Control, and Elimination Department, Pan American Health Organization, District of Colombia, United States of America

* perezf@paho.org

**Data Availability Statement:** Data is available within the manuscript.

## Abstract

*Histoplasma* antigen can be detected in people with advanced HIV disease (AHD), allowing for early and accurate diagnosis of histoplasmosis. The aim of this analysis was to assess the cost-effectiveness of routine histoplasmosis screening using antigen detection, among people with AHD. We developed a decision analytic model to evaluate *Histoplasma* antigen screening among people with AHD. The model estimated the costs, effectiveness, and cost-effectiveness of routine screening for *Histoplasma* antigen compared to the current practice of no routine *Histoplasma* antigen screening. The model includes stratification by symptoms of histoplasmosis, severity of presentation, and estimates of 30-day mortality. Data sources were taken from the Pan American Health Organization (PAHO) Strategic Fund databases on public purchases of medicines, and published literature on treatment outcomes. Outcome measures are life years saved (LYS), costs (US dollars), and incremental cost-effectiveness ratios (ICERs). Routine *Histoplasma* antigen screening avoids an estimated 17% of deaths in persons with advanced HIV disease, and is cost-effective compared to no histoplasmosis screening, with an ICER of $26/LYS. In sensitivity analysis assuming treatment for histoplasmosis with liposomal amphotericin, *Histoplasma* antigen screening remains cost-effective with an ICER of $607/LYS. *Histoplasma* antigen screening among people

**Funding:** This work was partially funded by the CDC-PAHO Agreement: Building Capacity and Networks to Address Emerging Infectious Diseases in the Americas (Award: 6 NU50CK000494-01-04; Funding Opportunity Announcement: CDC-RFA-CK18-1801CONT20). RR is supported by the National Institute of Allergy and Infectious Diseases (K23AI138851). The funders had no role in study design, data collection and analysis, decision to publish, or preparation of the manuscript.

**Competing interests:** The authors have declared that no competing interests exist.

with AHD is a cost-effective strategy and could potentially avert 17% of AIDS-related deaths. Prospective evaluation of histoplasmosis screening is warranted to determine effectiveness and treatment outcomes with this strategy.

## Introduction

Histoplasmosis is an important cause of mortality among people living with HIV [1–4]. The Joint United Nations Programme on HIV/AIDS (UNAIDS) estimates that in 2021, 2·5 million people were living with HIV and 34,000 AIDS-related deaths occurred in Latin America and the Caribbean region [5]. Among new HIV infections more than 30% present to care with advanced HIV disease (AHD; CD4 cell count of <200 cells/μL) [6]. These patients are at highest risk for opportunistic infections, including histoplasmosis [6, 7]. The number of deaths from HIV-associated histoplasmosis in Latin American countries is estimated to be on par with the number of deaths due to tuberculosis [1]. In highly endemic countries, histoplasmosis could represent the most common AIDS-defining condition [1].

*Histoplasma* antigen is detectable in urine, allowing for an early and accurate diagnosis. *Histoplasma* antigen testing has 95% overall sensitivity and 97% specificity [8, 9]. There is growing interest in screening all people with AHD for histoplasmosis to diagnose and treat patients earlier, thereby reducing AIDS-related mortality [6]. Studies from Guatemala and Paraguay have estimated the prevalence of *Histoplasma* antigen to be 8% and 10% in patients with AHD respectively [10, 11]. In this decision analytic model, we evaluate the cost-effectiveness of routine histoplasmosis screening among people living with AHD, compared to the current standard of care without routine histoplasmosis screening, where histoplasmosis diagnostics are only pursued if a patient is symptomatic.

## Methods

### Analytic overview

We developed a decision analytic model to evaluate histoplasmosis screening among people with AHD. Our model compared two strategies: 1) routine screening for histoplasmosis, a scenario in which all patients with AHD are screened for *Histoplasma* antigen independently of symptoms, and 2) no screening for histoplasmosis (the current standard of care), a scenario in which patients with AHD are not systematically screened, and diagnosis for histoplasmosis occurs only among symptomatic people and relies on conventional diagnosis methods triggered by a healthcare provider when there is clinical suspicion (**Fig 1**). We used the model to estimate costs, effectiveness, and cost-effectiveness of *Histoplasma* antigen screening strategies in Latin America and the Caribbean given the substantial burden of histoplasmosis in these regions [1]. The model includes stratification based on symptoms of histoplasmosis, severity of infection, and subsequent likelihood of mortality. Strategies were compared by direct healthcare costs and life years saved (LYS). Strategies were evaluated over a one month time horizon, with remaining life expectancy and healthcare costs included for patients who survived 30 days. The cost-effectiveness analysis was conducted from the perspective of the healthcare payer system, with costs and health outcomes discounted at 3% per year. Out of pocket costs incurred by patients such as transport, or loss of salary were not included. The perspective of the healthcare payer system was chosen as this would be most relevant to stakeholders considering investment in a histoplasmosis screening program. We calculated incremental cost-

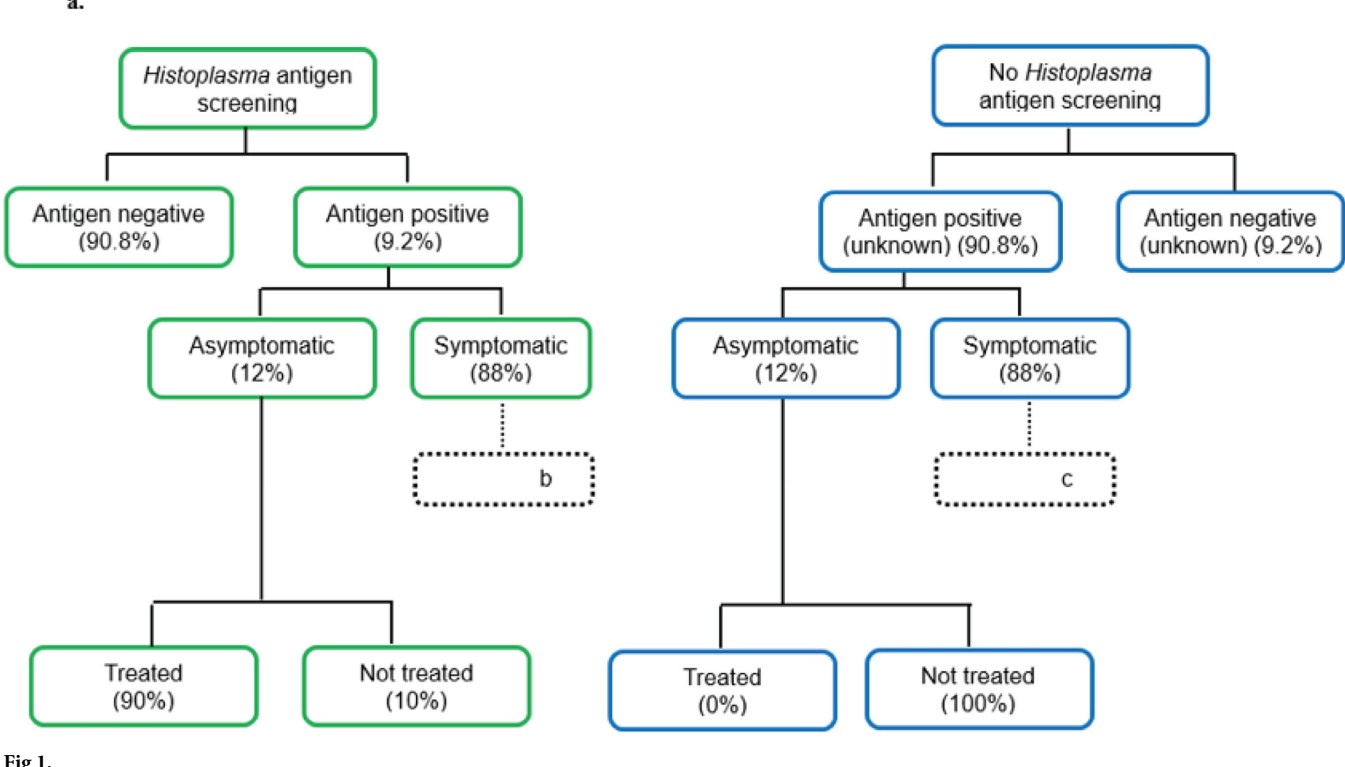

**Fig 1.**

effectiveness ratios (ICERs) of each strategy by dividing the additional cost by the additional LYS compared to the next less expensive strategy. The decision analytic model was implemented in Microsoft Excel version 2016.

## Diagnostic approach for histoplasmosis among people with AHD

Based on recent estimates, we assumed 293,426 people with AHD in Latin America and the Caribbean in 2020 [12]. In the scenario of routine histoplasmosis screening, screening was performed among all people with AHD. **Table 1** describes the input parameters and sources of data for the screening population. The model results presented as the "base case" use the parameter assumptions in **Table 1**.

Prior to any screening for histoplasmosis, there is a risk of death from other causes (other opportunistic infections) among all persons within the model. In the scenario of routine *Histoplasma* antigen screening, we assumed 80% received screening and returned for care. People with histoplasmosis were disaggregated into two categories: (1) asymptomatic; and (2) symptomatic (**Fig 1A**). Histoplasmosis screening was presumed to occur using a diagnostic test for *Histoplasma* antigen, which has 95% overall sensitivity (95% CI 94%-97%) and 97% specificity (95% CI 97%-98%) based on the accuracy of a real-life commercially available urine antigen test among persons with symptomatic histoplasmosis [8, 9]. Given the excellent performance of the *Histoplasma* antigen screening test, we did not account for false positive results in this model.

## Management of asymptomatic histoplasmosis

For asymptomatic people who have positive *Histoplasma* antigen testing, we assumed that all who returned for results were managed in outpatient services as mild to moderate

**Table 1. Description of advanced HIV disease population in the histoplasmosis screening and treatment model Latin America.**

| Population screened | Probability | Source |
|---|---|---|
| Risk of death among persons with CD4≤200 cells/μL from other causes over 90 days | 0.0353 | [27] |
| CD4≤200 cells/μL *Histoplasma* antigen screened | 0.8 | Assumption |
| Return to clinic for *Histoplasma* antigen results | 1 | Assumption |
| *Histoplasma* antigen prevalence | | |
| *Histoplasma* negative | 0·908 | [9, 10] |
| *Histoplasma* positive | 0·092 | [11, 28] |
| *Histoplasma* antigen positive, symptomatic | 0·88 | [29, 30] |
| Symptomatic hospitalized | 0·80 | Assumption |
| Among people screened and positive for *Histoplasma* antigen, conventional diagnostics performed | 0·05 | Assumption |
| *Histoplasma* antigen positive and conventional diagnostics positive | 0·77 | [6] |
| Absence of *Histoplasma* antigen screening, conventional diagnostics performed among symptomatic hospitalized | 0·60 | Assumption |
| Absence of *Histoplasma* antigen screening, conventional diagnostics performed among symptomatic non-hospitalized | 0·30 | Assumption |
| Symptomatic, known *Histoplasma* antigen positive, receives antifungal treatment | 0·90 | Assumption |
| Symptomatic, conventional diagnostics positive, receives antifungal treatment | 0·90 | Assumption |
| Symptomatic, conventional diagnostics negative, receives antifungal treatment | 0·20 | Assumption |
| Symptomatic, conventional diagnostics not performed, receives antifungal treatment | 0·30 | Assumption |
| Asymptomatic *Histoplasma* antigen positive | 0·12 | [29, 30] |
| Absence of *Histoplasma* screening, receives antifungal treatment | 0 | Assumption |
| Presence of *Histoplasma* screening, receives antifungal treatment | 0.9 | Assumption |
| **Histoplasmosis-positive outcomes** | | |
| Asymptomatic histoplasmosis outcomes | | |
| 30-day mortality if treated with antifungals | 0·10 | Assumption |
| 30-day mortality if not treated with antifungals | 0·10 | Assumption |
| Symptomatic histoplasmosis outcomes | | |
| 30-day mortality if treated with antifungals | 0·13 | [7, 11, 28] |
| 30-day mortality without antifungal treatment | 0·65 | [14] |

histoplasmosis; in these cases, WHO/PAHO guidelines recommend itraconazole 200 mg twice daily for 1 year [6]. The efficacy of itraconazole in asymptomatic histoplasmosis has not been widely studied. We modeled 30-day mortality of 10% for asymptomatic histoplasmosis positive people, with or without itraconazole.

## Management of symptomatic histoplasmosis

Among those patients with symptomatic histoplasmosis, 80% were presumed to be hospitalized with severe disease, and the remaining 20% were managed as outpatients with mild to moderate disease (**Fig 1B**). Among those who screened *Histoplasma* antigen positive, 5% had conventional histoplasmosis diagnostic testing performed, consisting of microscopy and fungal cultures. In the context of routine *Histoplasma* antigen screening, conventional diagnostics were less likely to be pursued, as patients with histoplasmosis would already have a positive antigen result. Conversely among people not antigen screened, we assumed 60% of those hospitalized had conventional histoplasmosis diagnostics performed, and 20% of outpatients had conventional diagnostics performed (**Fig 1C**). It was presumed that results of conventional diagnostics testing would take seven days.

Among those screened for *Histoplasma* antigen and symptomatic, we assumed 90% would receive treatment. The other 10% may be lost to follow-up or not engaged in medical care. Those hospitalized received amphotericin deoxycholate (1 mg/kg) for 14 days followed by 200mg of itraconazole twice daily, per the standard of care in Latin America [6, 13]. We assumed the average patient weighed 50kg, thus the amphotericin deoxycholate dose was 50mg per patient (1mg/kg). While the treatment for histoplasmosis is 14 days of amphotericin followed by itraconazole, we assumed that among those not screened for *Histoplasma* antigen, hospitalization was for 21 days, 7 days for conventional diagnostics to return with a definitive histoplasmosis diagnosis, and then 14 additional days for amphotericin treatment. Among people hospitalized, if conventional diagnostics were not pursued, their total hospital duration was 14 days for treatment. Symptomatic people who were *Histoplasma* antigen positive who were not hospitalized (20%) were treated as having mild to moderate disease with itraconazole (200 mg twice daily) for 12 months. Outpatients who received antifungal treatment were treated as mild to moderate histoplasmosis.

Overall, patients with histoplasmosis who received antifungal therapy had 13.7% 30-day mortality [11]. Untreated people were assumed to have 65% 30-day mortality [14]. This estimate is derived from a study from 1963 of untreated severe histoplasmosis, which identified 85% 30-day mortality [14]. We presumed reduced mortality from untreated histoplasmosis down to 65% (*expert opinion*) due to early diagnosis of HIV infection and widespread availability of antiretroviral therapy.

Among people not screened for *Histoplasma* antigen who were symptomatic, we assumed those with positive conventional diagnostics had an 90% probability of receiving treatment, those with negative diagnostics had a 20% probability of receiving treatment, and those without conventional diagnostics performed had a 30% probability of empiric treatment (**Fig 1C**). These estimates are based on expert opinion of clinicians from the Latin American region since there are no studies summarizing proportion of people with histoplasmosis who are treated in the context of conventional diagnosis.

## Health outcomes

Patients were assumed to have two primary outcomes: survival or death at 30 days. Lives saved (deaths avoided) was used as the main health outcome, reported as life years saved (LYS). We assumed the average age of those presenting to care was 38 based on the average age of histoplasmosis patients in Brazil and Guatemala [7, 11]. Average life expectancy for this age group was an additional 43 years–that is the average life expectancy from Brazil and Guatemala [15]. With a 3% discount rate, and 43 years of life lost from a death, 24·7 life years were lost per death, so 24·7 LYS were accrued per death prevented. Any patient alive at the end of 30 days was assumed to be cured of their histoplasmosis. These patients were assumed to have the average remaining life expectancy and incurred the costs of an additional 11 months of itraconazole. Other healthcare costs such as antiretroviral therapy and HIV monitoring labs were not incorporated into lifetime healthcare costs.

## Histoplasmosis screening and treatment costs

All costs were reported in 2022 US dollars (USD), and were assumed to have been borne fully by the Ministry of Health. Screening and treatment costs are reported in **Table 2**. A micro-costing approach was taken with conventional diagnostics, itraconazole treatment, and hospitalization-associated costs acquired from 10 sites, 5 in Brazil, 2 in Guatemala and one site each from Mexico, Colombia, and Argentina. From these sites, the mean of each cost was

**Table 2. Input costs of *Histoplasma* antigen screening and treatment.**

| | Average Cost (USD) | References, notes |
|---|---|---|
| *Histoplasma* antigen testing | $5·40 | [16] |
| *Histoplasma* microscopy and fungal culture | $56·19 | Micro-costing (see Methods) |
| Itraconazole 100 mg tablet | $0·28 | Micro-costing (see Methods) |
| Itraconazole 200 mg twice daily x 14 days | $15·68 | |
| Amphotericin deoxycholate 50 mg | $12·84 | [16] |
| Amphotericin 1mg/kg daily x 14 days | $179·76 | Assumes a 50 kg patient |
| **Hospitalization (21 days)** | | |
| Total 14-day treatment course: Amphotericin + Itraconazole | $195·41 | See above |
| Hospital stay | $1,386·77 | [31] |
| Laboratory testing | $62·25 | Micro-costing (see Methods) |
| Hospital supplies | $237·38 | Micro-costing (see Methods) |
| Chest x-ray | $11·38 | Micro-costing (see Methods) |
| Personnel | $27·50 | Micro-costing (see Methods) |
| **Total cost of hospitalization** | $1,920·68 | |
| Post hospitalization itraconazole | $380·80 | 11.5 months of itraconazole |
| Outpatient management of histoplasmosis | 35·06 | 30-day costs include laboratory monitoring, chest x-ray, clinic personnel. Excludes itraconazole. Micro-costing (see Methods) |

calculated, and used for the model. The cost of the *Histoplasma* antigen (EIA) was estimated at $5.40 per test as reported by the PAHO Strategic Fund [16].

Itraconazole costs of $0·28 USD per 100 mg tablet, or $15·68 USD for a 14-day course. The cost of 50 mg of amphotericin deoxycholate was $12·84 USD [16]. Assuming a 50 kg patient, a 14-day course would cost $179·76 USD. Hospitalization costs for histoplasmosis included hospital stay, laboratory testing (for monitoring while on amphotericin) hospital supplies (including intravenous fluids, potassium, and magnesium supplementation), one chest X-ray, and hospital personnel. Assuming a 21-day hospitalization, and 14 days of amphotericin-based treatment including laboratory monitoring and supplies, the total cost of hospitalization was $1,921 USD (**Table 2**). Among people who died within 30 days of histoplasmosis testing, we assumed all deaths occurred on day 19 [4].

Among people not hospitalized for histoplasmosis outpatient management costs included, laboratory testing, one chest x-ray, personnel for three visits, and itraconazole for 30 days. Total outpatient costs were estimated at $66 USD for one month. Among outpatients who were not treated with antifungals, we assumed one outpatient visit, thus accounted for personnel costs for one visit, outpatient laboratory testing, and one chest x-ray.

All people who were treated and survived beyond 30 days incurred the cost of 11 additional months of itraconazole.

## Definition of cost-effective

The primary outcome was the ICER, which is the incremental cost of screening vs. no screening, divided by the incremental effectiveness (LYS) of screening vs. not screening. While there is no strict definition of "cost-effective" for the purpose of our analysis, we used the WHO-CHOICE guidelines, which defines "cost-effective" as an ICER <3 times GDP per capita, and

"highly cost-effective" as an ICER < GDP per capita [17]. For reference, the Brazilian GDP per capita in 2021 was $7,519 USD, and the Guatemalan GDP per capita in 2021 was $5,026 USD. Realizing this is an imperfect definition, we also compared our analysis to the ICER of pre-exposure prophylaxis (PrEP) to prevent HIV infection in Brazil; this is an intervention that is widely accepted as cost-effective in Latin America. The results of a cost-effectiveness analysis found an ICER of PrEP compared to no PrEP of $2,530/LYS [18].

## Sensitivity analysis

As a sensitivity analysis, we modeled treatment with liposomal amphotericin, instead of amphotericin deoxycholate. The WHO/PAHO guidelines for the treatment of histoplasmosis disease suggests liposomal amphotericin (3 mg/kg) for fourteen days as first line therapy for hospitalized people [6]. We obtained costs of liposomal amphotericin 50mg (20mL vial) for seven Latin American countries, ranging from $112 to $494 per vial from the PAHO Strategic Fund Price Database (*personal communication*). We used the mean cost of $254·12 per vial. The average 50kg patient would require three vials per day for fourteen days. Thus, the total cost of 14 days of liposomal amphotericin would be $10,673 per patient. While liposomal amphotericin is recommended as first line therapy, it is not widely used due to the prohibitive cost.

We additionally explored the costs and outcomes of screening 100% of persons with advanced HIV disease for histoplasmosis. While this may not be realistic, this would provide the maximum efficacy possible with routine histoplasmosis screening. We performed one-way sensitivity analyses to evaluate the following possibilities: cost of *Histoplasma* antigen testing (at $20, $60, and $100 per test), *Histoplasma* antigen prevalence (ranging from 5% to 15%), and 30-day mortality among asymptomatic persons with histoplasmosis at 5%. We further explored the cost-effectiveness of *Histoplasma* antigen screening with varying mortality among symptomatic treated persons with histoplasmosis (ranging from 5% to 25% 30-day mortality), and among symptomatic untreated persons with histoplasmosis (ranging from 30% to 90% 30-day mortality).

## Role of the funding source

The study sponsor had no role in the study design, analysis, data interpretation, writing of the report, or decision to submit the paper for publication.

## Results

### Histoplasmosis screening in Latin America

In the base case model analysis using the assumptions in Tables 1 and 2, of 293,426 people with AHD in Latin America and the Caribbean, screening all people for *Histoplasma* antigen would cost $1·2 million. Without any *Histoplasma* antigen screening program 19,253 people would die at a cost of $33·7 million due to the cost of hospitalization and treatment of people with histoplasmosis.

In a scenario of systematic histoplasmosis screening and early diagnosis and treatment, the total cost of screening and treatment would be $36·0 million, with 15,839 deaths. The break-down of costs for screening and treatment are summarized in **Fig 2**. Thus, compared to no screening program, a *Histoplasma* antigen screening program across Latin America and the Caribbean would cost an additional $2·2 million, but with 17% (3,414) deaths avoided. The incremental cost per life saved is $648, or $26 per LYS (**Table 3**). **Fig 3** shows the comparison of costs with and without the *Histoplasma* antigen screening program. Given the GDP per

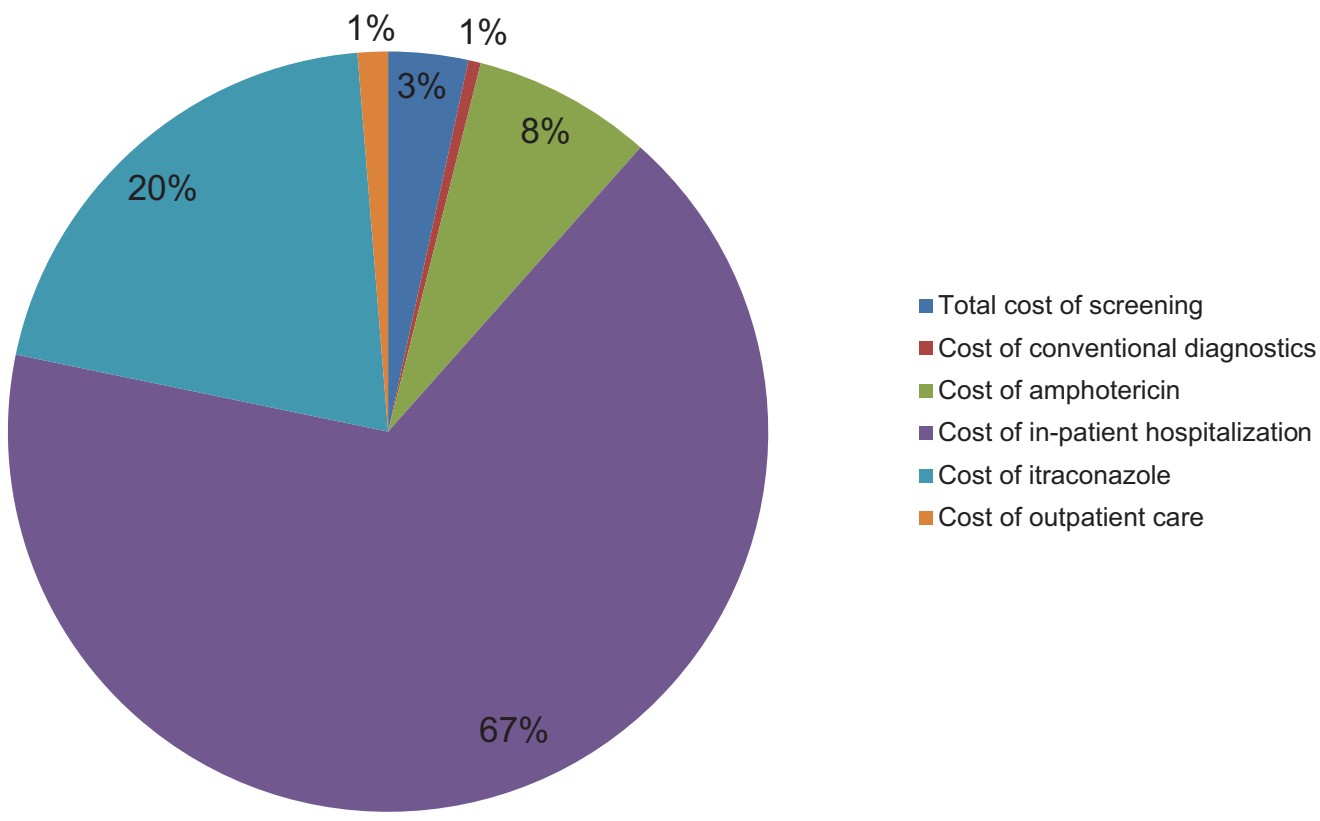

**Fig 2.**

capita specified in the methods, *Histoplasma* antigen screening would be considered highly cost-effective by international standards. Compared to other widely accepted interventions in Latin America, the ICER for *Histoplasma* antigen screening is approximately one tenth of the ICER of implementing PrEP to prevent HIV infection.

## Sensitivity analysis

In the absence of a *Histoplasma* antigen screening program the total cost of treating histoplasmosis with liposomal amphotericin would be $147 million dollars; 75% of those costs are due to the cost of liposomal amphotericin. With a screening program in place, the total cost of screening and treatment would be $198 million dollars. Thus, the incremental cost for a screening program would be $51 million dollars. Assuming equal efficacy as amphotericin deoxycholate the ICER with liposomal amphotericin would be $14,986 per life saved, or $607 per life year saved. While the ICER using liposomal amphotericin is 10-fold higher than the ICER using amphotericin deoxycholate, $607 per life year saved is still one tenth of the Guatemalan and Brazilian GDP per capita, and one quarter of the ICER that was acceptable for PrEP.

**Table 3. Cost-effectiveness results of *Histoplasma* antigen screening.**

|  | Cost (USD) | Incremental Cost | Effectiveness (life years) | Incremental effectiveness | ICER (Cost/LYS) |
|---|---|---|---|---|---|
| *No Histoplasma antigen screening* | $33,763,183 | -- | 423,567 | -- | -- |
| *Histoplasma antigen screening* | $35,975,763 | $2,212,580 | 507,886 | 84,319 | $26 |

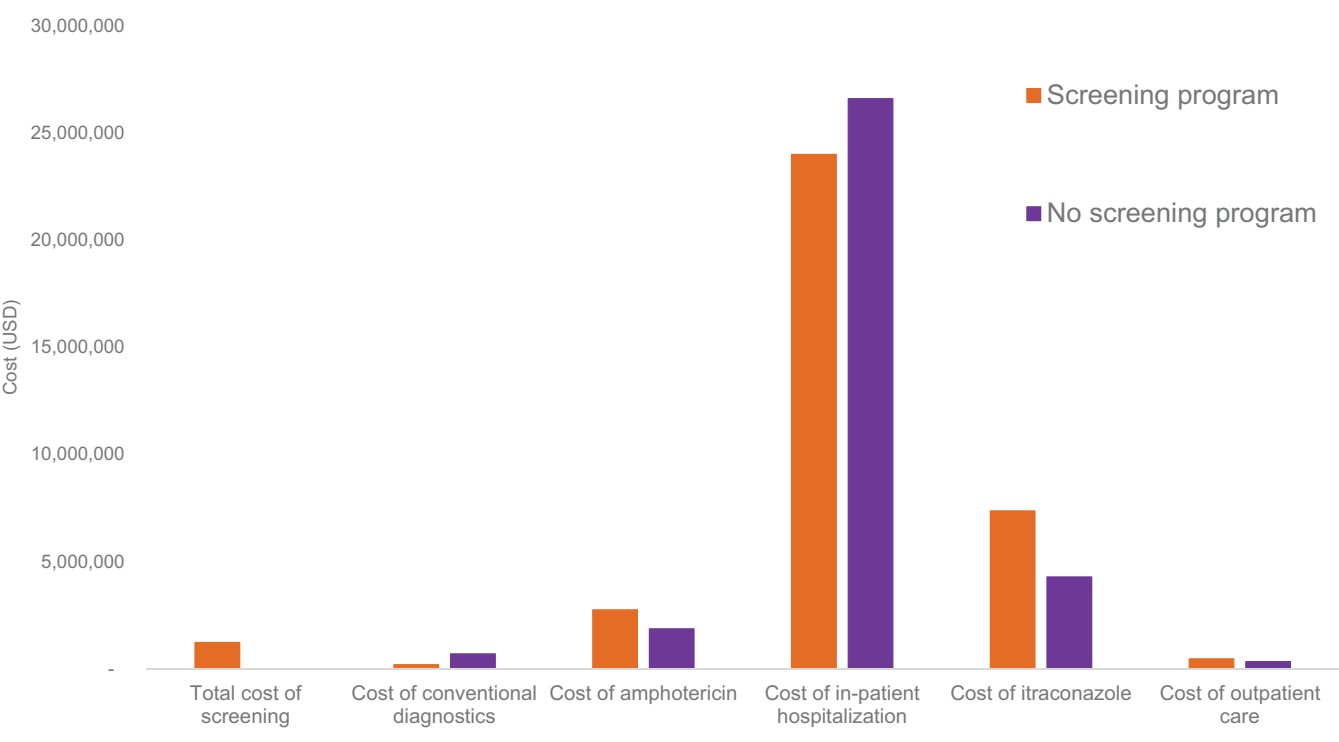

**Fig 3.**

When considering screening 100% of persons with advanced HIV disease for *Histoplasma* antigen, the total cost of screening and treatment would be 36·5 million. The incremental cost for a screening program would be 2.8 million with 20% of deaths avoided. The ultimate ICER is $648 per life saved, or $26 per life year saved. Assuming the *Histoplasma* antigen test cost $100 per test, the ICER of screening and treatment would be $280/LYS. Results of other sensitivity analyses are summarized in the Supporting Information (**S1–11 Tables**).

## Discussion

If *Histoplasma* antigen screening was widely implemented in Latin America and the Caribbean, such a screening program would cost $1·2 million. Early diagnosis with screening along with appropriate treatment would avert an estimated 17% of advanced HIV disease deaths, predominantly from early identification and treatment of persons with symptomatic histoplasmosis. This is the first analysis evaluating the potential cost-benefit of a histoplasmosis screening program, and we determined that a *Histoplasma* antigen screening program, with the current assumed costs and input parameters, would be considered highly cost-effective by international standards, even in low-income settings.

Current recommended treatment for severe and moderately-severe histoplasmosis is liposomal amphotericin [6]. However, access to this antifungal is limited in Latin America and the Caribbean due to its extremely high cost. Liposomal amphotericin is on the WHO Essential Medicine List, which generates new opportunities for advocacy for its introduction for improving management of histoplasmosis [19]. Liposomal amphotericin is less toxic and better tolerated compared to amphotericin deoxycholate. We estimated the unit cost of liposomal amphotericin was $254 per vial. This was the average cost of five Latin American centers, with a range from $112 to $493. Thus, for a 14-day course, the total cost of liposomal amphotericin

was >$10,000. This is unaffordable and as a result, the drug is infrequently used. Currently, a reduced price of $16·25 per vial of liposomal amphotericin is available only for leishmaniasis and cryptococcal meningitis [16]. Reduction of the price, like with an access program for LMICs or by introduction of generics and price negotiation with industry will increase the cost-benefit of liposomal amphotericin.

Limitations of our analysis are related to uncertainty with respect to model inputs, and model structure. For example, we assumed that 80% of people screened would return for *Histoplasma* antigen screening results. Without a point-of-care antigen test that yields timely results, there is likely some loss to follow up in people who are screened but do not return for results. We did not incorporate false positive antigen results into our model. We determined that with over 95% sensitivity and specificity, a minority would have false positives, and estimating the potential downstream effects of treating asymptomatic people with itraconazole would be inaccurate. Also, the high sensitivity and specificity assumptions are based on validation studies among persons with symptomatic histoplasmosis; it is possible that among asymptomatic persons, sensitivity and/or specificity may be lower. We assumed that those who survived the initial histoplasmosis infection would have the same survival of a person on ART as a person without HIV infection. In reality, social determinants of health may negatively affect life expectancy of PWH. Additionally, modeling cost-effectiveness for a continent means that there is variable *Histoplasma* prevalence, accessibility to antifungals, and costs. While overall our findings suggest that screening is cost-effective, there is likely variability by country and within country by region. More research is needed on prevalence of asymptomatic *Histoplasma* antigenemia, and outcomes of this population. Finally, we assumed there was access to antifungal medications, which is variable in many settings in Latin America.

This analysis highlights many areas of uncertainty and future research. We considered a general *Histoplasma* antigen screening test, and estimated an average cost of $5·40, as this is the cost of the EIA. However, there are advances in the development of a *Histoplasma* antigen point-of-care lateral flow assay (LFA) that has shown a high sensitivity and specificity, and if widely available would improve access to rapid results [20]. It is necessary for countries to give priority access to high-quality diagnostics for the rapid detection of *Histoplasma* antigen. As a first step, the *Histoplasma* antigen test was recently included in the WHO list of Essential Diagnostics [21]. Innovation, price negotiation, and pooled procurement mechanisms are critically needed to improve access, especially where histoplasmosis is highly endemic as has been shown in Latin America [1].

The burden of histoplasmosis by country and region is needed to further understand the impact of histoplasmosis on AIDS-related mortality. Such studies would better determine the ideal screening test for histoplasmosis and the optimal cost of such a test. The performance of antigen tests among asymptomatic persons with advanced HIV disease would clarify real world sensitivity and specificity in the screening population. Few studies have described clinical outcomes of people with histoplasmosis who are not hospitalized. Many histoplasmosis cases might be misdiagnosed as smear negative tuberculosis and result in poor clinical outcomes [11, 22, 23]. Additionally, among people with AHD, co-infection with tuberculosis, cryptococcosis, and/or pneumocystis pneumonia, is possible [11, 22]. Prevalence, diagnosis, and management of co-infections among people with histoplasmosis is an area of much needed research.

In cryptococcal meningitis a single dose of liposomal amphotericin was recently found to be non-inferior to the traditional seven day course of amphotericin with flucytosine [24]. Similarly, a phase II randomized clinical trial is underway evaluating short course liposomal amphotericin for people with disseminated histoplasmosis and AIDS [25]. If proven safe and effective, this short course could reduce toxicity, adverse events, and hospitalization duration

and thereby costs of treatment of histoplasmosis. Finally, implementation of histoplasmosis screening, diagnosis, and treatment needs further study in order to be integrated into the existing healthcare system [26].

In conclusion, we determined that *Histoplasma* antigen screening in Latin America and the Caribbean would avert an estimated 3,414 deaths, at a cost of $648 USD per life saved. By international standards, such a screening program is highly cost-effective for low- and middle-income countries. The present estimates are based on current prices, but price negotiation and pooled procurement mechanisms for *Histoplasma* diagnostics and treatments could enhance access and could lead to even greater cost-benefit.

## Supporting information

**S1 Table. Sensitivity analysis results evaluating cost-effectiveness of *Histoplasma* antigen screening with treatment with liposomal amphotericin.**
(DOCX)

**S2 Table. Sensitivity analysis results evaluating cost-effectiveness of *Histoplasma* antigen screening with treatment with 100% screened.**
(DOCX)

**S3 Table. Sensitivity analysis results evaluating cost-effectiveness of *Histoplasma* antigen testing cost of $20 per test.**
(DOCX)

**S4 Table. Sensitivity analysis results evaluating cost-effectiveness of *Histoplasma* antigen testing cost of $60 per test.**
(DOCX)

**S5 Table. Sensitivity analysis results evaluating cost-effectiveness of *Histoplasma* antigen prevalence of 5%.**
(DOCX)

**S6 Table. Sensitivity analysis results evaluating cost-effectiveness of *Histoplasma* antigen prevalence of 15%.**
(DOCX)

**S7 Table. Sensitivity analysis results evaluating cost-effectiveness of *Histoplasma* antigen if 30-day mortality for asymptomatic histoplasmosis is 5% (treated or untreated).**
(DOCX)

**S8 Table. Sensitivity analysis results evaluating cost-effectiveness of *Histoplasma* antigen if 30-day mortality for symptomatic histoplasmosis is 5% (treated).**
(DOCX)

**S9 Table. Sensitivity analysis results evaluating cost-effectiveness of *Histoplasma* antigen if 30-day mortality for symptomatic histoplasmosis is 25% (treated).**
(DOCX)

**S10 Table. Sensitivity analysis results evaluating cost-effectiveness of *Histoplasma* antigen if 30-day mortality for symptomatic histoplasmosis is 30% (untreated).**
(DOCX)

**S11 Table. Sensitivity analysis results evaluating cost-effectiveness of *Histoplasma* antigen if 30-day mortality for symptomatic histoplasmosis is 90% (untreated).**
(DOCX)

## Acknowledgments

Centers that provided the data:

1. Guatemala, Guatemala -Unidad de Atención Integral Clinica Familiar "Luis Angel Garcia"/ Hospital General Sana Juan de Dios- Eduardo Arathoon;

2. Guatemala City, Guatemala-Unidad de Atención Integral del VIH e Infecciones Crónicas- "Dr. Carlos Rodolfo Mejía Villatoro"/Hospital Roosevelt-Johana Samayoa;

3. Buenos Aires, Argentina -Hospital "Juan A. Fernández"—Sección Microbiología, Servicio Laboratorio de Análisis Clínicos-Liliana Guelfand;

4. Porto Alegre, Brazil-Hospital de Clínicas de Porto Alegre (HCPA)-Diego Falci;

5. Porto Alegre, Brazil-Grupo Hospitalar Conceição (GHC)-Andressa Noal;

6. Porto Alegre, Brazil- Associação Hospitalar Vila Nova (AHVN)-Nicole Reis;

7. Porto Alegre, Brazil- Santa Casa de Misericordia de Porto Alegre—Alessandro C. Pasqualotto; 8. Campo Grande, Brazil—Divisão de Doenças Infecciosas, Universidade Federal de Mato Grosso do Sul-Anamaria Miranda;

9. Rio de Janeriro, Brazil -Laboratório de Micologia, Instituto Nacional de Infectologia Evandro Chagas–Fundação Oswaldo Cruz-Rosely Zancope;

10. México City, México -Department of Infectious Diseases, Laboratory of Clinical Microbiology, Instituto Nacional de Ciencias Médicas y Nutrición Salvador Zubirán, Mexico City, Mexico-Rosa Martinez;

11. Medellin, Colombia -Unidad de Micología Medica y Experimental, Corporación para Investigaciones Biológicas (CIB)—Alejandra Zuluaga. Global Action For Fungal Infections (GAFFI)- Juan Luis Rodriguez Tudela and Ana Alastruey-Izquierdo.

**Disclaimer:** The findings and the conclusions in this report are those of the authors and do not necessarily represent the views of the Centers for Disease Control and Prevention.

## Author Contributions

**Conceptualization:** Radha Rajasingham, Alexander Jordan, Omar Sued, Tom Chiller, Freddy Perez.

**Data curation:** Radha Rajasingham, Narda Medina, Gabriel T. Mousquer, Diego H. Caceres, Mathieu Nacher, Diego R. Falci, Ayanna Sebro, Alessandro C. Pasqualotto, Omar Sued.

**Formal analysis:** Radha Rajasingham, Alessandro C. Pasqualotto.

**Funding acquisition:** Freddy Perez.

**Investigation:** Radha Rajasingham, Narda Medina, Diego H. Caceres, Omar Sued, Tom Chiller, Freddy Perez.

**Methodology:** Radha Rajasingham, Narda Medina, Gabriel T. Mousquer, Diego H. Caceres, Alexander Jordan, Mathieu Nacher, Diego R. Falci, Ayanna Sebro, Alessandro C. Pasqualotto, Omar Sued, Freddy Perez.

**Resources:** Mathieu Nacher, Diego R. Falci, Freddy Perez.

**Supervision:** Diego H. Caceres, Alessandro C. Pasqualotto, Tom Chiller, Freddy Perez.

**Validation:** Narda Medina, Mathieu Nacher, Diego R. Falci, Ayanna Sebro, Omar Sued, Tom Chiller, Freddy Perez.

**Writing – original draft:** Radha Rajasingham.

**Writing – review & editing:** Radha Rajasingham, Narda Medina, Diego H. Caceres, Alexander Jordan, Mathieu Nacher, Diego R. Falci, Ayanna Sebro, Alessandro C. Pasqualotto, Omar Sued, Tom Chiller, Freddy Perez.

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
