## [Decision Letter · Decision Letter 0]

22 Mar 2023

PGPH-D-23-00160

Cost-effectiveness evaluation of routine histoplasmosis screening among people living with advanced HIV disease in Latin America and the Caribbean

Dear Dr. Rajasingham,

Thank you for submitting your manuscript to PLOS Global Public Health. After careful consideration, we feel that it has merit but does not fully meet PLOS Global Public Health’s publication criteria as it currently stands. Therefore, we invite you to submit a revised version of the manuscript that addresses the points raised during the review process.

The analysis is on an important topic. I share the methodological concerns identified by one of the reviewers. Some of them (i.e., sensitivity analyses and presentation of results) can be addressed easier than others (i.e., choice of the modeling approach). I would like to invite you to address these aspects during a major revision with an emphasis on a more detailed analysis regarding the major uncertainty in assumptions and a further discussion of the limitations regarding competing risks of other comorbidities.

We look forward to receiving your revised manuscript.

Kind regards,

Stefan Kohler, Ph.D., M.D.

Academic Editor

Journal Requirements:

1. Please send a completed 'Competing Interests' statement, including any COIs declared by your co-authors. If you have no competing interests to declare, please state "The authors have declared that no competing interests exist". Otherwise please declare all competing interests beginning with the statement "I have read the journal's policy and the authors of this manuscript have the following competing interests:"

b. If any authors received a salary from any of your funders, please state which authors and which funders.

3. We notice that your supplementary table 1 is included in the manuscript file. Please remove them and upload them with the file type 'Supporting Information'. Please ensure that each Supporting Information file has a legend listed in the manuscript after the references list.

4. In the online submission form, you indicated that "Data will be made available upon request". All PLOS journals now require all data underlying the findings described in their manuscript to be freely available to other researchers, either 1. In a public repository, 2. Within the manuscript itself, or 3. Uploaded as supplementary information.

Additional Editor Comments (if provided):

Reviewers' comments:

Reviewer's Responses to Questions

**Comments to the Author**

1. Does this manuscript meet PLOS Global Public Health’s publication criteria? Is the manuscript technically sound, and do the data support the conclusions? The manuscript must describe methodologically and ethically rigorous research with conclusions that are appropriately drawn based on the data presented.

Reviewer #1: Yes

Reviewer #2: Partly

2. Has the statistical analysis been performed appropriately and rigorously?

Reviewer #1: Yes

Reviewer #2: N/A

3. Have the authors made all data underlying the findings in their manuscript fully available (please refer to the Data Availability Statement at the start of the manuscript PDF file)?

Reviewer #1: Yes

Reviewer #2: Yes

4. Is the manuscript presented in an intelligible fashion and written in standard English?

Reviewer #1: Yes

Reviewer #2: Yes

5. Review Comments to the Author

Reviewer #1: Line 138. In order to better understand the paper, it is important to specify the year of the estimate of 256,585 people with HIV in Latin America.

Lines 155- 157

183-187

Check the consistency between points 155 to 157 and 183 to 187. Also, record the parameters to consider a reduction of 85 to 65% of mortality

Reviewer #2: Rajasingham et al present an interesting analysis that uses simulation modeling to project the cost-effectiveness of using histoplasmosis antigen screening among people with advanced HIV disease (AHD) in Latin America and the Caribbean. This is an important question that is well-suited to be addressed by simulation modeling. The authors have determined some important data sources for several parameters. However, there are a number of assumptions made throughout the parameterization of the decision tree (see Table 1). In particular, the assumption that false positives will not be influential needs additional discussion and testing in the model; especially given that data are not provided about the test characteristics of urinary histo antigen for asymptomatic people. Assumptions are also made about 100% of people taking the test and acting upon the results, which is not realistic and will overestimate the benefits of a screening program. Additionally, data have been used from histoplasmosis infection among people without HIV without further discussion regarding whether outcomes would be the same among PWH. To address many of these concerns, the authors could perform one-way sensitivity analyses on parameters to examine the impact of a range of estimates for these assumptions. This is critical to include as per ISPOR-SMDM recommendations. Assessment of multi-ways sensitivity analyses for parameters influential in one-way sensitivity analysis would also be important to include.

Additionally, it is not clear that a decision tree is the best modeling approach to use. Because people with AHD are at high risk for a number of opportunistic infections and have extremely high mortality, it is a major assumption that treating people for histoplasmosis will result in long-term survival, as immunosuppressed people with HIV (CD4<200/uL) may will experience a high risk of death from TB, serious bacterial infection, or other diseases. Competing risks of deaths from other highly prevalent comorbidities are not captured in the current approach and are likely to reduce the benefits of screening for one of these comorbidities (i.e., histoplasmosis). The authors also make the assumption that people with AHD who are treated for histoplasmosis will live for the average life expectancy of people in Brazil; however, people living with HIV in Brazil are at increased risk of experiencing disadvantageous social determinants of health and therefore may be less likely to live to the national average life expectancy. The decision tree model does not account for lifetime morbidity/mortality risks among PWH, costs of ART, loss to follow-up from ART and HIV care. Therefore, the extremely attractive estimate of $26/YLS for Histoplasmosis screening in this high-risk population does not include lifetime costs of HIV care or the potential for a shorter life expectancy.

The results are presented as a population finding, as opposed to a per person costs and outcomes, which is not the traditional approach. Importantly, the total costs are presented as the total costs of the program, but do not include any of the other HIV care costs that would be essential if people with HIV survive after screening/treatment for histoplasmosis. Inclusion of these important costs will reduce the cost-effectiveness of the intervention because people who survive will cost more per year than is currently included.

A small query is that most modeling analyses that use published data are still reviewed by IRB to confirm that they meet "exempt" criteria.

I am concerned that the analysis as current presented does not accurately portray the full costs or examine the potential cost-effectiveness of such a screening program.

6. PLOS authors have the option to publish the peer review history of their article (what does this mean?). If published, this will include your full peer review and any attached files.

**Do you want your identity to be public for this peer review?** For information about this choice, including consent withdrawal, please see our Privacy Policy.

Reviewer #1: **Yes: **Gloria Aguilar

Reviewer #2: No

---

## [Editor Report · Decision Letter 1]

28 Apr 2023

PGPH-D-23-00160R1

Cost-effectiveness evaluation of routine histoplasmosis screening among people living with advanced HIV disease in Latin America and the Caribbean

Dear Dr. Perez,

Thank you for submitting your manuscript to PLOS Global Public Health. After careful consideration, we feel that it has merit but does not fully meet PLOS Global Public Health’s publication criteria as it currently stands. Therefore, we invite you to submit a revised version of the manuscript that addresses the points raised during the review process.

Before sending out the revised manuscript for review, I would like you to consider again the following comments:

Main comment

R2/sensitivity analyses: I concur with the reviewer and remain interested in the results of further sensitivity analyses on the modelling assumptions. Additional information on the dependency of the presented findings with respect to the chosen model parameters could strengthen the study. Even with a lack of data on the plausible variation of parameters, I would find a comprehensive sensitivity analysis informative, as it helps assess the robustness and determinants of the presented findings. I would like to encourage you to perform and include a comprehensive (one-way) sensitivity analysis as supplementary material and incorporate the main results of this analysis in the main text.

Minor comments

R1/Line 138: Please add the year/period in addition to the referenced study.R2/HIV care: Consider incorporating the response in the manuscript as background information.R2/population finding: Consider presenting your chosen evaluation perspective, motivation for it, and implications of the evaluation perspective in the methods.Consider aligning the formulation for the headings "Treatment for asymptomatic histoplasmosis-positive people" and "Management of symptomatic histoplasmosis" to emphasize that these sections cover the complementary aspects.It seems that only some, but not all revisions were tracked in version with track changes. Please resubmit a clean version and a version in which all text changes are tracked. Consider accepting formatting changes to create a version with tracked changes.

We look forward to receiving your revised manuscript.

Kind regards,

Stefan Kohler, Ph.D., M.D.

Academic Editor
---

## [Decision Letter · Decision Letter 2]

25 Jun 2023

PGPH-D-23-00160R2

Cost-effectiveness evaluation of routine histoplasmosis screening among people living with advanced HIV disease in Latin America and the Caribbean

Dear Dr. Perez,

Thank you for submitting your manuscript to PLOS Global Public Health. After careful consideration, we feel that it has merit but does not fully meet PLOS Global Public Health’s publication criteria as it currently stands. Therefore, we invite you to submit a revised version of the manuscript that addresses the points raised during the review process.

Please revisit the past reviewer comments and revise the manuscript with respect to substantially strengthen in the examination of the impact of data uncertainty on the model projections, including, among others, sensitivity analyses test characteristics (what are the risks of false positives/false negatives?), cure (what if the treatment is less effective than base case estimates?), mortality if untreated (what if risks of death are actually worse?), efficiency of screening (only a % of tests are linked to care).

We look forward to receiving your revised manuscript.

Kind regards,

Stefan Kohler, Ph.D., M.D.

Academic Editor

Journal Requirements:

Additional Editor Comments (if provided):

Reviewers' comments:

Reviewer's Responses to Questions

**Comments to the Author**

1. If the authors have adequately addressed your comments raised in a previous round of review and you feel that this manuscript is now acceptable for publication, you may indicate that here to bypass the “Comments to the Author” section, enter your conflict of interest statement in the “Confidential to Editor” section, and submit your "Accept" recommendation.

Reviewer #1: All comments have been addressed

Reviewer #2: (No Response)

2. Does this manuscript meet PLOS Global Public Health’s publication criteria? Is the manuscript technically sound, and do the data support the conclusions? The manuscript must describe methodologically and ethically rigorous research with conclusions that are appropriately drawn based on the data presented.

Reviewer #1: Yes

Reviewer #2: No

3. Has the statistical analysis been performed appropriately and rigorously?

Reviewer #1: Yes

Reviewer #2: N/A

4. Have the authors made all data underlying the findings in their manuscript fully available (please refer to the Data Availability Statement at the start of the manuscript PDF file)?

Reviewer #1: Yes

Reviewer #2: Yes

5. Is the manuscript presented in an intelligible fashion and written in standard English?

Reviewer #1: Yes

Reviewer #2: Yes

6. Review Comments to the Author

Reviewer #1: I have no additional comment

Reviewer #2: This revised manuscript remains an interesting use of modeling to assess an important public health question. Although this revision now includes 3 sensitivity analyses, this seems to be an insufficient examination of the uncertainty in input estimates. If such a strong cost-effectiveness conclusion is to be supported by modeling evidence, then there needs to be a much more rigorous examination of the impact of data uncertainty on model projections.

7. PLOS authors have the option to publish the peer review history of their article (what does this mean?). If published, this will include your full peer review and any attached files.

**Do you want your identity to be public for this peer review?** For information about this choice, including consent withdrawal, please see our Privacy Policy.

Reviewer #1: **Yes: **Gloria Celeste Aguilar Barreto

Reviewer #2: No

---

## [Editor Report · Decision Letter 3]

5 Jul 2023

Cost-effectiveness evaluation of routine histoplasmosis screening among people living with advanced HIV disease in Latin America and the Caribbean

PGPH-D-23-00160R3

Dear Dr Perez,

We are pleased to inform you that your manuscript 'Cost-effectiveness evaluation of routine histoplasmosis screening among people living with advanced HIV disease in Latin America and the Caribbean' has been provisionally accepted for publication in PLOS Global Public Health.

Best regards,

Stefan Kohler, Ph.D., M.D.

Academic Editor